# Echocardiographic Findings in Canine Model of Chagas Disease Immunized with DNA *Trypanosoma cruzi* Genes

**DOI:** 10.3390/ani10040648

**Published:** 2020-04-09

**Authors:** Olivia Rodríguez-Morales, Francisco-Javier Roldán, Jesús Vargas-Barrón, Enrique Parra-Benítez, María de Lourdes Medina-García, Emilia Vergara-Bello, Minerva Arce-Fonseca

**Affiliations:** 1Department of Molecular Biology, National Institute of Cardiology “Ignacio Chávez”, Juan Badiano No. 1, Col. Sección XVI, Tlalpan, Mexico City 14080, Mexico; rm.olivia@gmail.com (O.R.-M.); benitez_PE@hotmail.com (E.P.-B.); sayaca24@hotmail.com (M.d.L.M.-G.); emilyvbct@hotmail.com (E.V.-B.); 2Department of Echocardiography, National Institute of Cardiology “Ignacio Chávez”, Juan Badiano No. 1, Col. Sección XVI, Tlalpan, Mexico City 14080, Mexico; roldan@cardiologia.org.mx (F.-J.R.); ecovarjes@gmail.com (J.V.-B.)

**Keywords:** echocardiography, Chagas disease, *Trypanosoma cruzi*, canine model, DNA vaccine

## Abstract

**Simple Summary:**

Chagas disease (ChD) has become one of the biggest public-health problems in Latin America due to its incapacitating effects and mortality rates. Therefore, it is important to detect cardiac involvement as early as possible in order to estimate the risk and prognosis before the patient becomes symptomatic. Development of anti-*Trypanosoma cruzi* vaccines could significantly contribute to the control of ChD. The aim of this study was to examine the usefulness of echocardiography in determining the prophylactic effect of the DNA vaccines on heart damage using *T. cruzi* genes cloned into an expression vector, which was intramuscularly injected to beagle dogs. Structural and functional changes in the chagasic heart were monitored easily by echocardiography, and it was determined that plasmid DNA vaccination with *T. cruzi* genes induced moderate protection in immunized dogs avoiding enlargement of cardiac chambers, poor contractibility, and heart failure, especially with pBCSP plasmid (the construct with *TcSP* gene of *T. cruzi*, encoding *trans*-sialidase protein), as was reported previously.

**Abstract:**

Chagas disease (ChD) is considered an emerging disease in the USA and Europe. *Trypanosoma cruzi* genes encoding a *trans*-sialidase protein and an amastigote-specific glycoprotein were tested as vaccines in canine model. The aim for this study was determining the prophylactic effect of these genes in experimentally infected dogs by echocardiography evaluation to compare with our findings obtained by other techniques published previously. Low fractional-shortening values of non-vaccinated dogs suggested an impairment in general cardiac function. Low left ventricular ejection fraction values found in infected dogs suggested myocardial injury regardless of whether they were vaccinated. Low left ventricular diastolic/systolic diameters suggested that progressive heart damage or heart dilation could be prevented by DNA vaccination. Systolic peak time was higher in non-vaccinated groups, increasing vulnerability to malignant arrhythmias and sudden death. High left ventricular volume suggested a decrease in wall thickness that might lead to increased size of the heart cavity, except in the pBCSP plasmid-vaccinated dogs. There was an echocardiographic evidence of left ventricular dilation and reduction in systolic function in experimental chagasic dogs. Echocardiography allowed a more complete follow-up of the pathological process in the living patient than with other techniques like electrocardiography, anatomopathology, and histopathology, being the method of choice for characterizing the clinical stages of ChD.

## 1. Introduction

Chagas disease (ChD) is caused by the protozoan parasite *Trypanosoma cruzi*, which infects humans and more than 100 species of domestic and sylvatic mammals and can be transmitted by over 150 species of hemiptera insects of the subfamily Triatominae (Reduviidae) [1,2,3,4]. ChD has become one of the biggest public-health problems in Latin America due to its incapacitating effects and mortality rates [5,6,7]. According to data reported by the World Health Organization, Argentina, Brazil, and Mexico are the three countries with the highest estimated number of infected people. Estimated numbers of chagasic cardiopathy cases are highest in Argentina, Brazil, Colombia, Bolivia, and Mexico [1]. Nevertheless, patterns of emigration from Chagas-endemic areas have drastically altered the epidemiology of this disease in the United States, Europe, and other non-endemic regions in recent decades, making it one of the neglected tropical diseases now found in non-endemic areas of the world [8,9].

There are two successive stages in ChD: acute and chronic phases. In the acute phase, cardiac involvement may occur in up to 90% of cases. After six to eight weeks, most patients show recovery of the clinical manifestations [9]. In the chronic phase of ChD, during which parasites are hidden in target tissues, especially the cardiac and digestive system muscles, different clinical forms may be observed: (i) the asymptomatic form; (ii) the cardiac form, which occurs in around 30% of the patients with disorders of the heart’s electrical conduction system, arrhythmia, heart-muscle disorder, heart failure (HF), or secondary embolisms; (iii) the digestive form, with localized lesions and enlargement of the oesophagus and the colon; and (iv) a mixed form (cardiac plus digestive) that affects around 10% of patients. Patients ultimately die, usually from sudden death caused by arrhythmias or HF, which often occurs in early adulthood [1]. The main causes of death associated with chronic Chagas’ cardiomyopathy (CCC) are progressive congestive HF and sudden cardiac death [10].

The diagnosis of ChD can be made by serology a few weeks after the primary infection. Large-scale epidemiological studies have consistently demonstrated that about three-quarters of seropositive patients remain asymptomatic throughout their entire life [11]. It is therefore important to detect cardiac involvement as early as possible in order to estimate the risk and prognosis before the patient becomes symptomatic [12]. Cardiac damage is suspected by ≥1 of the following electrocardiographic findings: right bundle-branch block, left anterior fascicular block, atrioventricular blocks, multiform ventricular beats, and sinus bradycardia [13].

In chronic Chagas heart disease, the electrocardiogram (EKG) has been the method of choice for detecting myocardial damage, especially in remote endemic areas, because of its low cost, portability. and simplicity. For noninvasive detection of early myocardial damage, other methods have been used, including Holter monitoring, echocardiography, nuclear scans, and stress tests [12]. Two-dimensional and tissue Doppler imaging echocardiography are useful, complementary methods for the follow-up of patients with chronic ChD myocarditis. Both techniques provide valuable information on cardiac structure and function that complements information provided by EKG, allowing the recognition of left ventricular (LV) systolic and diastolic dysfunction, right ventricle involvement, and regional contractility abnormalities, including typical apical aneurysms. The cardiac ultrasound has utility in the diagnosis, classification, and detection of early myocardial damage and the prognostic assessment of patients with ChD [8,13,14,15].

Dogs play a major role in the domestic cycle of *Trypanosoma cruzi*, acting as reservoirs, but they can also develop acute and chronic disease, similar to human infection; therefore, canine Chagas’ disease has become a major veterinarian concern in the Americas [16,17,18]. In small animals’ veterinary medicine, echocardiography is a common technique performed on dogs to noninvasively assess systolic myocardial function. This technique has been shown to be repeatable and reproducible in dogs [19]. In addition, the canine model has gained wide acceptance as another experimental model to study a wide variety of conditions associated with ChD; however, the development of vaccines as a prophylactic method has not been widely addressed [20,21,22,23,24,25,26,27,28,29,30].

Development of anti-*T. cruzi* vaccines could significantly contribute to the control of ChD. Immunological protection against experimental infection with *T. cruzi* has been studied since the second decade of the last century, testing many types of immunogens [31]. In our previous studies, *TcSP* (encoding *trans*-sialidase protein) and *TcSSP4* (encoding amastigote-specific protein) genes were evaluated prophylactically as DNA vaccines in both murine and canine model of ChD [32,33,34,35,36]. The clinical, serological, and postmortem parameters obtained were consistent among these studies, showing a moderate effectiveness in protection against experimental ChD. The effect of these genes used as vaccines on the clinical parameters was evaluated through general physical examinations as a consequence of immunization and through their effect on the cardiac protection determined by electrocardiography [34]. However, these studies do not include complementary premortem examinations about both structural and functional cardiac abnormalities provided by medical imaging methods such as echocardiography, which could be useful in obtaining accurate data on the development of morphological changes in the heart of experimentally infected and immunized/infected chagasic dogs.

The aim of this study was to examine the usefulness of echocardiography in determining the prophylactic effect of the DNA vaccines on heart damage using *T. cruzi* genes cloned into an expression vector, which was intramuscularly injected to beagle dogs. To do so, we use this cardiac imaging method to measure hemodynamic parameters to determine whether echocardiography provides accurate data on the development of morphological changes in the heart of experimentally infected and immunized/infected chagasic dogs and if these data are consistent with those reported previously [34,35,36].

## 2. Materials and Methods

### 2.1. Experimental Animals

Animal handling and experimental procedures were approved by the Bioethics Committee of the National Institute of Cardiology Ignacio Chávez and performed under the established guidelines of the International Guiding Principles for Biomedical Research involving Animals and the Norma Oficial Mexicana (NOM-062-ZOO-1999) Technical Specifications for the Care and Use of Laboratory Animals [37]. Forty-six male and female beagle dogs aged 16 (± 1) months and weighing 13.79 (± 3.72) kg were used in this study. The beagle dogs were purchased from Criadero El Atorón (Teotihuacán, Estado de México; Mexico) without distinction of sex. Animals were housed in the appropriate Animal Facility, maintained on a 16 h/8 h light-dark cycle under conditions of controlled temperature (20–22 °C) and humidity, and fed a standard commercial dry food formulated for dogs, and water were available *ad libitum*. All dogs had a completed schedule of preventive medicine. A standardized enzyme-linked immunosorbent assay (ELISA) [38] used to diagnose ChD in the dogs ruled out any previous infection by *T. cruzi*.

### 2.2. Immunizations

The *T. cruzi* genes encoding the *Tc*SSP4 (amastigote-specific protein) and *Tc*SP (*trans*-sialidase protein) antigens were cloned in the commercially available eukaryotic expression vector pBK-CMV (Stratagene) to generate two constructs, pBCSSP4 and pBCSP, respectively, as described previously [34]. The dogs were immunized with doses of 500 g DNA dissolved in 500 L saline solution (SS) twice at 15-day intervals by intramuscular injection between the semitendinosus and semimembranosus muscles of the pelvic limbs using a 3-mL syringe with a 21 G × 32 mm needle. The dogs were separated in seven experimental groups (A–G) (Table 1) in order to compare the disease progression after DNA-vaccine administration.

### 2.3. Trypanosoma cruzi Challenge of Dogs

The dogs immunized with DNA or mock-immunized with SS as well as three animals belonging to positive control group were challenged intraperitoneally two weeks after the last immunization with 5 × 10^5^ metacyclic trypomastigotes per animal, which were obtained from urine and feces of triatomes and resuspended in SS of a well-characterized Mexican *T. cruzi* Ninoa strain (MHOM/MX/1994/Ninoa (*T. cruzi*)) [39,40]. The rest of the positive control dogs (*n* = 6) were infected with an inoculum size that ranged from 50 × 10^3^ to 2 × 10^6^ metacyclic trypomastigotes. The amount of *T. cruzi* Ninoa strain inoculum had no effect on the severity of the canine ChD [34,35,36,38].

### 2.4. Echocardiography

Transthoracic echocardiography (Phillips IE33) with a 2.5–3.5 MHz probe was performed in all dogs three times at 1-week intervals before start the study and during the chronic stage of infection (12 months after inoculation in vaccinated or chronically infected groups) in order to detect and compare morphological changes in the dogs’ heart among basal and/or reference values. The animals were positioned in dorsal and right lateral decubitus without any chemical restriction during the study; this could be achieved with previous training through daily manipulation in examination tables in order to keep the dogs in the desired position to carry out the study. In those few cases which chemical restriction was necessary (only 5 dogs and exclusively in their basal records), acepromazine (Calmivet ^®^ tablets) at light sedation dose of 1.25 mg/kg of body weight was used by oral administration. Image acquisition was done by means of a right parasternal long-axis view and right apical two- and four-chamber view in bidimensional mode according to Boon’s method [41]. Heart rate was simultaneously calculated from the preceding R-to-R interval on the electrocardiogram. End-diastolic and end-systolic diameters as well as end-diastolic and end-systolic free wall thickness of the left ventricle were measured. LV septum walls, LV end-diastolic volume, and the systolic peak time were also calculated. Parameters of LV systolic function, i.e., fractional shortening (FS %), and left ventricular ejection fraction (LVEF %) calculated from the end-diastolic and end-systolic volumes in the standard four-chamber long-axis 2D-echo views were also recorded with the following formula ((end-diastolic volume − end-systolic volume)/end-diastolic volume) × 100.

### 2.5. Euthanasia

At the end of the echocardiographic study, at 12 months postinfection, chronic chagasic dogs were euthanized to perform other studies *postmortem* such as macroscopic evaluation and histology in order to determine cardiac damage [30]. Briefly, the euthanasia method indicated for dogs was used according to the Norma Oficial Mexicana (NOM-033-ZOO-1995) Humane Sacrificing of Domestic and Wild Animals [42] through intravenous injection of barbiturate derivatives at a dose of 30 mg/kg and then a lethal dose of 15% potassium chloride administered intravenously. This protocol was approved by the Committee for the Care and Use of Animals of Laboratory (CICUAL, for its acronyms in Spanish) and the Bioethics Committee of the National Institute of Cardiology Ignacio Chávez (Registration number: 08-578) and performed under the established guidelines of the International Guiding Principles for Biomedical Research involving Animals.

### 2.6. Statistical Analysis

All data obtained (continuous variables with a normal distribution determined by Kolmogorov–Smirnov test) were analyzed with commercially available software IBM^®^ SPSS Statistics version 22.0 (NY, USA). The parametric statistical test of one-way ANOVA was used to examine the vaccination effect within each group for echocardiographic variables. When a significant difference was detected, the Tukey test was used to compare the means. Results were expressed as mean ± standard deviation. The level of significance in all tests was set at *p* ≤ 0.05.

## 3. Results

### 3.1. Infection Establishment 

The experimental *T. cruzi* infection was confirmed microscopically (parasitemia) in all infected groups by examining freshly isolated blood samples collected from the brachiocephalic vein every third day, observing from 200 to 400 parasites/mL as limit of detection intermittently between day 22 and day 55 postinfection. *T. cruzi* infection was diagnosed by the ELISA method two months postinoculation and was confirmed by indirect immunofluorescence technique in all unimmunized/infected dogs. All animals were monitored clinically by general physical examinations and electrocardiographic studies [34].

### 3.2. Vaccination Ameliorated the LVEF

To correlate the prophylactic effects of DNA vaccines with *T. cruzi* genes reported previously on the protection of heart damage with the reduction of the cardiac function alterations by hemodynamic parameters, echocardiography was performed. This noninvasive method provided the useful tool to evaluate both anatomy structure and function of the heart. The mean ± standard deviations values of some cardiovascular parameters recorded at each group (specified in Table 1) are showed in Table 2.

The infected dogs (B group) had fractional shortening (FS %) values significantly reduced, very similar to those obtained in mock-immunized (G group) and empty cloning vector-immunized (F group) dogs, suggesting an impairment in general cardiac function since their basal values showed percentages within the reported reference values [41] and similar to those that were always recorded in the healthy control group (Table 2).

Significant differences in the LVEF values among infected and noninfected dogs could be demonstrated regardless of whether they were vaccinated, although the lowest values belonged to the group nonimmunized with *T. cruzi* DNA (G group) (Table 2). No significant differences could be demonstrated in the LVEF values among immunized and nonimmunized infected dogs, although slightly higher values in D and E groups (pBCSP-immunized/infected and pBCSSP4-immunized/infected dogs, respectively) could be detected (Table 2), suggesting a better effectiveness of pumping into the systemic circulation.

### 3.3. Plasmid DNA Immunization Prevented the Increase in LV Diastolic and Systolic Diameter

Both LV diastolic and systolic diameter were also significantly high in B, F, and G groups (infected, empty cloning vector-immunized/infected, and SS mock-immunized/infected, respectively), while the values observed in vaccinated dogs were slightly lower (*p* < 0.05) than those of dogs that did not receive either vaccination in which progressive heart damage or heart dilation was diagnosed (Table 2).

### 3.4. Vaccination Protected Against Arrhythmias and Sudden Death

Neither infection nor the vaccination/infection altered the heart rate. However, systolic peak time was significantly higher in infected (B group) and mock-vaccinated/infected (G group) dogs (Table 2); conversely, immunization with *T. cruzi* genes or with the empty plasmid maintained this parameter as healthy or reference values (Table 2), showing less vulnerability to malignant arrhythmias and sudden death.

### 3.5. TcSP Gene Avoided Left Ventricular Dilation

LV volume was also elevated in all infected dogs (B and E–G groups) (Table 2), suggesting a decrease in wall thickness that might lead to increased size of the heart cavity; on the other hand, values in this parameter in animals vaccinated with the plasmid that carried the *TcSP* gene (D group) resemble those obtained in the healthy control (A group) and immunized/noninfected (C group) dogs (Table 2).

### 3.6. Summary of Previous Published Results

About the protection of the vaccine at the clinical level, it was observed that the hyperthermia of the acute phase was avoided, that lymphadenopathy was detected, that tachycardia was prevented in both acute and chronic stages, and that the severity and number of abnormalities on the EKG decreased [34] (Table 3). All infected/unimmunized dogs exhibited parasitemia starting on day 22 and lasting until day 55 postinfection, whereas the parasitemia of immunized/infected dogs occurred over a shorter time period, from day 32 to 46 postinfection [34] (Table 3). The effect of vaccination with these plasmids on macroscopic and microscopic damage caused by the chronic *T. cruzi* infection was to prevent splenomegaly and to avoid serious damage to the heart, in which lesions were limited to the subepicardial tissue [36] (Table 3).

## 4. Discussion

In this study, echocardiographic features found in experimental infected and DNA-immunized/infected dogs in chronic phase of the ChD were obtained. These features provided accurate data about cardiac structure and function, which complemented information given by other methods previously reported by us (Table 3) [34,36]. As expected, most of typical abnormalities found in these chagasic dogs by EKG studies, histology, and *postmortem* examination could be seen in nonimmunized/infected dogs, and they are in concordance with the evidence from this imaging technique.

A low FS percentage was found in nonimmunized dogs. This parameter is an index of general cardiac function that can provide a comprehensive evaluation of left ventricular systolic function with other recommended echocardiographic parameters. When FS is low such as in infected, pBK-CMV vaccinated, and SS-mock immunized groups it may be secondary to poor preload, increased after load, or decreased contractility [41]. There are some studies that report systematic tests for right ventricular dysfunction by FS evaluation. Right ventricular function is an important predictor of mortality and quality of life in patients with LV failure, myocardial infarction, congenital heart disease, and pulmonary hypertension [43,44]; according to this, we suggest that non-vaccinated dogs developed an ability reduction of the myocardial fibers to distend, which together with the diastolic dimensions suggests little contractility. It is documented that a high afterload hindering muscle contraction could show a decreased FS as an important feature by echocardiography [43], such as that we can observe in those non-vaccinated/infected dogs, suggesting an impairing in general cardiac function.

The LVEF percentage had a decrease in all experimental infected groups, suggesting significant myocardial injury caused by *T. cruzi* infection. This finding is consistent with a study performed with a group of 89 patients with CCC classified according to the presence of normal or pseudo-normal ventricular filling pattern; there were some with pseudo-normal filling pattern who reported a significant LV systolic impairment in terms of LV dimensions, wall motion score, and LVEF [45].

Diastolic measurements were higher in the groups that were not immunized with any *T. cruzi*-DNA vaccine than in those of the healthy control dogs and of the immunized dogs. Specifically, they were higher in the infected animals (B group), in those which were immunized with pBK-CMV plasmid (F group), and in the SS-mock immunized (G group). This parameter assesses size of cardiac chambers; if the parameter is high, it reflects a volume increase inside the chamber and therefore heart dilation [41]. However, among the groups that received the immunizations, only pBCSP-immunized/infected animals had a low volume inside the chamber similar than healthy control dogs; therefore, this demonstrated that *T. cruzi* gene encoding *Tc*SP *trans*-sialidase protein is able to protect against heart dilation or to avoid progressive heart damage if it is used as a prophylactic measure. Also, the vaccination could be effective to avoid more severe disorders such as malignant arrhythmias and sudden death because, in accordance with Biolo et al. [15], the LV systolic dysfunction in ChD may be present immediately after an early parasympathetic dysautonomia, which is a condition where autonomic derangements enhance the dependency of cardiac output increase on volume and shape modifications, requiring more ventricular dilation and forceful contraction and also triggering microcirculatory vasospasm, another important mechanism in Chagas cardiomyopathy.

LV systolic diameter high values in animals of B, F, and G groups in the present study, which did not receive DNA immunization as preventive action, showed a marked cardiac function damage demonstrated by a significant increase of LV systolic diameter as well as systolic peak time. The systolic measurements by echocardiography assess cardiac function as well as the increased LV chamber size during systole [41]. Two-dimensional, tissue Doppler, and strain echocardiography have been used in dogs and cats to assess both left and right ventricular function in normal and pathological conditions to estimate intracardiac pressures and myocardial dysfunction, to diagnose cardiomyopathies, and to assess inter- and intraventricular synchrony [46].

The LV volume remained in reference values in those animals receiving the *TcSP* gene as DNA vaccine (D group), suggesting that pBCSP immunization may protect in developing an increase in ventricular mass and end-diastolic volume as adaption mechanism by a chronic volume overload [47] like hypertrophy and marked dilation commonly seen in CCC. Those animals that showed increased LV volume could be developing left atrial dilation, which is an indicator of the severity of volume overload and increased cardiac pressures. This feature is one of the most important prognostic factors for humans, dogs, and cats with heart disease [46]. In addition, this finding was in accordance with our previous results [36], which showed that macroscopic cardiac alterations like whitish areas in the heart of fibrous consistency, abundant pericardial fluid, LV hypertrophy, thinning of the right ventricular wall, severe tricuspid endocarditis, and heart adherences with trachea and pericardium were absent in all dogs vaccinated with pBCSP plasmid (Table 3).

The overall evaluation of these echocardiographic parameters indicates that the hearts of nonimmunized/infected dogs had volume overload with FS % below (27.32–31.64%) the references values (33–46%) [41] and, together with the increase of the systolic dimensions, can be inferred that those animals with experimental chagasic infection that did not receive any immunizations developed certain degree of HF. This is important because CCC may be detected with or without symptoms. Most investigators combine clinical and EKG findings, cardiomegaly, and systolic dysfunction observed by echocardiography into four groups of progressive heart damage. The recent American College of Cardiology/American Heart Association staging of disease progression classifies the CCC into A (high risk of HF without structural heart disease), B (structural heart disease without HF), C (structural heart disease with prior or present HF), and D (refractory HF) [48]. Asymptomatic subjects comprise about three quarters of seropositive persons, and those with a normal EKG are referred as being in the indeterminate phase of the disease (stage A). About 2% to 18% of patients had been classified in stage B, which no cardiomegaly is present and LV systolic function is normal. Symptomatic patients with mild to moderate cardiac damage (stage C and/or D) may present arrhythmias, embolism, sudden death, and reversible HF, as well as dilated heart and abnormal LV systolic and diastolic functions; others may have LV apical and other segmental wall abnormalities [13]. Diagnosis by echocardiography in our experimental dogs was a suitable tool to detect structural heart disease that would be able to classify the chagasic patients into A or C stages for the infected ones and into B or D for the vaccinated-infected dogs, even being asymptomatic animals. This classification in these dogs can be possible because the more severe cardiac abnormalities registered by echocardiography were in the infected dogs than in immunized-infected dogs. Therefore, it is possible to assume that vaccination helped to avoid the highest level of damage in the heart that lead to HF.

This study, using dogs as an experimental animal model suggests that more valuable clinical data could be obtained by echocardiography for the management of the patient than those found by electrocardiography or by evidence of cardiac damage observed postmortem at necropsy or by histopathology. However, our echocardiography recordings were performed from 12 months after the challenge with *Trypanosoma cruzi*, being a study limitation. Further research should consider using echocardiography at an earlier stage of the ChD in order to follow up the pathological process, to give a timely symptomatic treatment, or to even try the trypanocidal therapy because, although acute chagasic myocarditis is infrequent, appearing in only 1% to 5% of those having the acute phase (1 to 5 of every 10,000 infected subjects), there are published echocardiographic series on ChD patients who had abnormal two-dimensional echocardiograms in 52% and pericardial effusion in 42%. In addition, mean-LVEF was normal (63%) in these patients. Apical or anterior dyskinesis was found in 21%, and only 6% had LV dilation. These findings demonstrate the need to perform echocardiograms to rule out other cardiac conditions to lead to HF and to evaluate LV dysfunctions during the acute phase of ChD [13].

Echocardiography has been very useful in prognosis of some canine diseases. For example, heart disease is not well characterized in adult dogs with Duchenne muscular dystrophy, so Guo et al. defined the clinical course of cardiomyopathy by evaluating echocardiography and cardiac magnetic resonance imaging in 24 golden retriever dogs. The study showed that dogs with muscular dystrophy have a progressive cardiomyopathy characterized by LV systolic and diastolic dysfunction, LV chamber enlargement, myocardial tension abnormalities, and myocardial fibrosis similar to that seen in dilated cardiomyopathy (DCM) [49]. The purpose of another study was to investigate the prognostic value of several clinical variables, EKG, and echocardiography in 63 dogs with idiopathic DCM; it was concluded that, in dogs with DCM, the restrictive transmitral flow pattern seems to represent a useful prognostic indicator. HF degree, ascites, end-systolic volume index, and LVEF are also useful parameters if an adequate transmitral flow pattern is not recorded [50]. Piantedosi’s group studied 120 dogs using standard echocardiography to assess the effect of loading condition changes on LV volumes before and 24 h after the patent ductus arteriosus occlusion by Amplatzer Canine Duct Occluder. After ductal closure, a significant reduction of all the parameters examined could be seen: LV internal diameter at end-diastole; LV internal diameter at end-systole; end-diastolic and systolic volume; end-diastolic volume index; and FS. Twenty-four hours after closure, the evaluation of the relative percentage difference of the selected echocardiographic parameters showed a significant reduction, higher in the small size breed dogs than in large size ones [51]. Although echocardiographic analysis at varying canine diseases shows a diagnosis of DCM, epidemiological data will always provide the guideline to the veterinarian in those cases where ChD is suspected; the gold standard in this case is the serological diagnosis.

There is a growing literature demonstrating the potential of nonconventional echocardiography in clinical and subclinical cardiac conditions; speckle tracking echocardiography is capable to discriminate between different causes of LV hypertrophy by ultrasound deformation imaging [52]. However, several limitations should be acknowledged before seeking to try new techniques. The lack of some normal reference ranges in veterinary echocardiography precluded the suitable interpretation of values obtained. Nevertheless, our main objective was to compare a few imaging parameters among healthy dogs and immunized/infected and nonimmunized/infected dogs with findings previously obtained by other diagnostic non-imaging studies.

## 5. Conclusions

In conclusion, it has successfully demonstrated that ChD in experimental infected dogs resulted in cardiac dilation and that plasmid DNA vaccination with *T. cruzi* genes induces moderate protection in immunized dogs, avoiding enlargement of cardiac chambers, poor contractibility, and HF, especially with pBCSP plasmid. Using echocardiography, structural and functional changes in the chagasic heart were monitored easily and without pain or discomfort to the patient. Also, this method was a suitable tool to evaluate the protection of progressive heart damage or heart dilation provided by the prophylactic effect of the DNA vaccine. Transthoracic echocardiography should be the method of choice for characterizing the clinical stages of ChD, such as systolic dysfunction during the acute phase of ChD, and to rule out a rapidly treatable cause of HF (e.g., pericardial effusion).

## Figures and Tables

**Table 1 animals-10-00648-t001:** Study design for *T. cruzi* experimentally infected dogs immunized with DNA vaccine containing the genes encoding a *trans*-sialidase protein (pBCSP) or an amastigote-specific glycoprotein (pBCSSP4).

Group	Group Description and *n*	Vaccine	Postinfection (PI) or Postvaccination (PV) Time for the Echocardiographic Study
A	Nonimmunized/Noninfected (healthy control) *n* = 5	None	NA
B	Chronically infected control *n* = 9	None	12 months PI
C	Each plasmids-immunized/Noninfected *n* = 6	pBCSP (*n* = 2) pBCSSP4 (*n* = 2) pBK-CMV (*n* = 2)	12.5 months PV
D	Immunized with *TcSP* gene/Infected *n* = 9	pBCSP	12 months PI
E	Immunized with *TcSSP4*/Infected *n* = 9	pBCSSP4	12 months PI
F	Immunized with cloning vector/Infected *n* = 4	pBK-CMV	12 months PI
G	Mock-immunized with saline solution/Infected *n* = 4	Saline Solution	12 months PI

NA = Not applicable, because they were neither infected nor vaccinated; pBCSP = plasmid carrying *TcSP* gene; pBCSSP4 = plasmid carrying *TcSSP4* gene; pBK-CMV= empty cloning vector.

**Table 2 animals-10-00648-t002:** Cardiovascular parameters in *T. cruzi* experimentally infected dogs immunized with DNA vaccine containing the genes encoding a *trans*-sialidase protein (pBCSP) or an amastigote-specific glycoprotein (pBCSSP4).

Group	Fractional Shortening (FS) (%)	Left Ventricular Ejection Fraction (LVEF) (%)	Left Ventricular (LV) Diastolic Diameter (mm)	Left Ventricular (LV) Systolic Diameter (mm)	Heart Rate (bpm)	Systolic Peak Time (ms)	Left Ventricular Volume (mL)
**A** **(healthy control)** **(Non-imm/Non-inf)**	39.29 ± 3.76	77.60 ± 4.22	28.70 ± 2.84	17.42 ± 1.98	101.80 ± 24.14	212.40 ± 20.11	18.56 ± 3.11
**B** **(Inf)**	31.64 ± 8.12 *	50.29 ± 6.59 *	34.19 ± 7.15 *	23.35 ± 5.83 *	102.87 ± 21.58	261.93 ± 31.57 *	33.97 ± 18.62 *
**C** **(P-imm/Non-inf)**	**41.19 ± 4.13**	**78.20 ± 5.17**	**26.30 ± 2.30**	**15.44 ± 1.46**	**128.60 ± 31.38**	**189.00 ± 15.51**	**16.80 ± 1.30**
**D** **(pBCSP-imm/Inf)**	**33.57 ± 6.31**	54.62 ± 9.19 *	**29.90 ± 2.43**	**20.68 ± 2.77**	**93.00 ± 11.17**	**206.25 ± 15.31**	**23.73 ± 6.01**
**E** **(pBCSSP4-imm/Inf)**	**33.94 ± 9.17**	58.45 ± 12.03 *	**30.06 ± 1.88**	**20.84 ± 2.36**	**112.63 ± 26.38**	**193.00 ± 31.92**	25.49 ± 5.18 *
**F** **(pBK-CMV-imm/Inf)**	31.55 ± 1.13 *	52.33 ± 5.37 *	31.70 ± 34.59 *	21.67 ± 2.80 *	137.67 ± 30.66	244.00 ± 19.16	32.83 ± 4.66 *
**G** **(SS mock-imm/Inf)**	27.32 ± 8.29 *	47.67 ± 4.93 *	30.2 ± 1.13 *	21.80 ± 4.81 *	159.67 ± 37.53 *	240.33 ± 16.74 *	29.13 ± 9.02 *

All data are expressed as means and standard deviations. Statistical significances (*p* ≤ 0.05) are assigned as * when there was a significant difference with group A control healthy dogs and/or with reference values [41]. Bold values belong to the groups which were vaccinated and did not show significant difference with group A control healthy dogs. bpm = beats per minute, imm = immunized, inf or Inf = infected, P-imm = each plasmids-immunized, pBCSP = plasmid carrying *TcSP* gene, pBCSSP4 = plasmid carrying *TcSSP4* gene, pBK-CMV= empty cloning vector, and SS = saline solution.

**Table 3 animals-10-00648-t003:** Other findings obtained previously in the same *T. cruzi* experimentally infected dogs immunized with DNA vaccine containing the genes encoding a *trans*-sialidase protein (pBCSP) or an amastigote-specific glycoprotein (pBCSSP4).

Group (Description)	Electrocardiographic Abnormalities (Percentage of Animals with These Alterations) [34,36]	Parasitemia [34]	*Postmortem* Macroscopic Alterations (Percentage of Animals with These Findings) [36]	Cardio- and/or Splenomegaly [36]	Histopathology [36]
**A (healthy control)**	None	Negative	None	None	None
**B (Inf)**	Ischemia and conduction block LV hypertrophy (100%)	Positive at 22–55 days postinfection	Whitish areas in heart of fibrous consistency and abundant pericardial fluid (69%)	Cardiomegaly	Inflammatory lesions from subepicardium, myocardium, and subendocardium
**C (P-imm/Non-inf)**	None	Negative	None	None	N/D
**D (pBCSP-imm/Inf)**	AV Block (33%) and LV enlargement (11%)	Positive at 32–46 days postinfection	None	Cardiomegaly	Inflammatory lesions from subepicardium and myocardium
**E (pBCSSP4-imm/Inf)**	MIMI (11%) and Second-degree AV block (11%)	Positive at 32–46 days postinfection	Heart with adhesions in trachea and pericardium, splenic thickened walls and whitish areas in spleen and heart (11%)	None	Inflammatory lesions from subepicardium
**F (pBK-CMV-imm/Inf)**	LV enlargement (75%) and Right BBB (25%)	Positive at 31–46 days postinfection	Abundant pericardial fluid and whitish areas in spleen (75%). Ascites, megaesophagus, LV hypertrophy, thinning of the RV wall, and severe tricuspid endocarditis (25%)	Cardiomegaly Splenomegaly	Inflammatory lesions from subepicardium, myocardium, and subendocardium
**G (SS mock-imm/Inf)**	VPC (25%), pericardial effusion (50%), myocardial infarction and/or pericarditis, and RV enlargement (50%)	Positive at 21–55 days postinfection	Whitish areas in heart of fibrous consistency and abundant pericardial fluid (75%)	Cardiomegaly Splenomegaly	Inflammatory lesions from subepicardium, myocardium, and subendocardium

imm = immunized, inf or Inf = infected, P-imm = each plasmids-immunized, pBCSP = plasmid carrying *TcSP* gene, pBCSSP4 = plasmid carrying *TcSSP4* gene, pBK-CMV= empty cloning vector, and SS = saline solution. LV = left ventricular, AV = atrioventricular, MIMI = microscopic intramural myocardial infarctions, BBB = bundle branch block, VPC = ventricular premature complexes, RV = right ventricular, N/D = not determined.

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
