# Peer review of "Echocardiographic Findings in Canine Model of Chagas Disease Immunized with DNA Trypanosoma cruzi Genes"

_animals, 2020, doi:10.3390/ani10040648_

Round 1

Reviewer 1 Report

The study has been carried out to asses the higher quality of echocardiography in the respect of electrocardiogram, to detect miocardial damage by Trypanosoma cruzi infection and damages mitigation after putative vaccine.

The study has been developed rigorously and the paper is well written with correct English form.

The only doubt is: is not this evidence already known, in general? it is really necessary to organize all the complex study, involving many resources and domestic animal model to produce this evidence?

Anyway, the paper is accepted for publication and the only notes is that, in Results, tab 2 must be recalled in each results paragraph, to be avoid confusion and that, probably the Table 3 must be inserted in the results paragraph also if results are obtained previously.

Author Response

April 2nd, 2020.

Dear reviewer:

We respectfully submit revision of our article, titled: Echocardiographic findings in canine model of Chagas disease immunized with DNA Trypanosoma cruzi genes, for your kind consideration. The article is submitted for the special section: Veterinary Clinical Studies, Special Issue Animal Models for the Study of Cardiovascular Disease, at MDPI Animals journal.

We are thankful to you and the another reviewer for providing the constructive feedback that has helped us to improve the presented contents. All points raised by the reviewers are addressed in this version, and we have provided point-by-point response to reviewer comments.

We look forward to hearing a positive response from you.

With warm regards

Minerva Arce-Fonseca PhD.

Researcher in Medical Sciences

Responsible of Laboratory of Molecular Immunology and Proteomics

Department of Molecular Biology

Instituto Nacional de Cardiología Ignacio Chávez

Mexico City, Mexico

RESPONSE TO REVIEWERS

All changes were made using “Track Changes” function in Microsoft Word as Assistant Editor suggested. We used the version our manuscript found at the above link for our revisions provided by the editorial office.

Reviewer #1: Thank you for finding that the article had been developed rigorously and is well written with correct English form. We appreciate that you find the article suitable for publication in MDPI Animals. Thank you for your positive feedback.

The only doubt is: is not this evidence already known, in general?

Not really. A lot is known about the cardiac effects of the parasite; however, in this study DNA vaccines were evaluated whose effect was not known, these genes had not been tested before. As you know, each antigen produces a different immune response and that response may be different in each animal model (de Lana, 2017).

Our study aims to contribute to offering prophylactic options against Chagas disease. As it is well known, there are no vaccines for human protection, much less for domestic dogs.

The results of these vaccines in relation to clinical damage, immunological effect and target tissue damage, which have already been published as independent articles (Rodríguez-Morales et al. 2012 and 2013; Arce-Fonseca et al. 2013), did not demonstrate that any of them monitor the progress of the disease in the living patient, if perhaps the electrocardiographic study by providing evidence of the electrical conduction of the heart and indirectly functional damage.

Evaluating the structural and functional changes in the chagasic heart based on echocardiography was important because the intention of a vaccine, among other things, is to prevent the progression of the disease until it reaches heart failure. The best tool to do this is echocardiography. With this echocardiographic study, which greatly complemented what was already known with other studies, we were able to know that vaccination with plasmid DNA with T. cruzi genes induced moderate protection in immunized dogs.

-de Lana, M. Experimental studies of Chagas disease in animal models. In American Trypanosomiasis Chagas Disease One Hundred years of Research, J. Telleria and M. Tibayreng, Ed., Elsevier, Netherlands, Amsterdam, 2017, pp. 299-320,

It is really necessary to organize all the complex study, involving many resources and domestic animal model to produce this evidence?

We understand your concern, and share the idea of applying the three Rs in the management of laboratory animals: Replacement, Reduction and Refinement, in order to use the least amount of resources and animals; this study was carefully designed for this purpose. As you know, this kind of researches (vaccine evaluations) is really necessary to organize all the complex study to produce this evidence. Definitively, the evaluation of the effect of a vaccine must be investigated in its different clinical phases, and before reaching them, the use of evidence in laboratory animals is necessary. In the case of Chagas disease, which does not have a vaccine currently available, animal models to evaluate the many candidates that different research groups are developing are very valuable; even more the canine model on the murine one.

Anyway, the paper is accepted for publication and the only notes is that, in Results, table 2 must be recalled in each results paragraph, to be avoid confusion and that, probably the Table 3 must be inserted in the results paragraph also if results are obtained previously.

Thank you for pointing this out. We have edited recalled Table 2 in each results sentence to avoid confusion. About Table 3 must be inserted in the results paragraph, we added a new 3.5 sub-section of results as you suggested.

Reviewer 2 Report

This Msc deals with the use of Echocardiographic examination during a chagas disease infection in dogs, immunized or not.

This technic could help to monitor the clinical stages and evolution of te disease.

I thik it is of great value in this field.

I would suggest only a few minor revisions following:

Introduction

Line 43: cite a review in which the different vertebrate and invertebrate hosts are reported.

Line 45 : cite an article in which ChD is reported as the biggest public-health problems in Latin Americ

Material and method:

I think the Table I is very clear, so sub-section 2.3 is useless.

From line 157 to 163 : This is results. And it is missing in the result part.

Line 183 « ] » is missing at the end of the formula.

Statistical analyzes : Did authors test the normality prior to use the One Way ANOVA test ? If yes, please indicate which test was done. If no, please test the normality and change to non-parametric test instead of One-way ANOVA if the datas do not follow a normal distribution.

Results

I would be curious to have correlation tests between the results of echocardiography and the parasitic load. Maybe, the vaccination leaded to parasitic load decrease and so a diminution of the effect on heart ? I recommend to discuss about other diseases which could lead to the same echocardiographic profile. They could interfere in the first results and also in the following of the disease. This point should also be discussed in discussion section.

Discussion

I recommend to discuss about other diseases which could lead to the same echocardiographic profile. They could interfere in the first results and also in the following of the disease.

Author Response

April 2nd, 2020.

Dear reviewer:

We respectfully submit revision of our article, titled: Echocardiographic findings in canine model of Chagas disease immunized with DNA Trypanosoma cruzi genes, for your kind consideration. The article is submitted for the special section: Veterinary Clinical Studies, Special Issue Animal Models for the Study of Cardiovascular Disease, at MDPI Animals journal.

We are thankful to you and another reviewer for providing the constructive feedback that has helped us to improve the presented contents. All points raised by the reviewers are addressed in this version, and we have provided point-by-point response to your comments.

We look forward to hearing a positive response from you.

With warm regards

Minerva Arce-Fonseca PhD.

Researcher in Medical Sciences

Responsible of Laboratory of Molecular Immunology and Proteomics

Department of Molecular Biology

Instituto Nacional de Cardiología Ignacio Chávez

Mexico City, Mexico

RESPONSE TO REVIEWERS

All changes were made using “Track Changes” function in Microsoft Word as Assistant Editor suggested. We used the version our manuscript found at the above link for our revisions provided by the editorial office.

Reviewer #2: Thank you for finding that the manuscript is of great value in this field. We appreciate that you find the article suitable for publication in MDPI Animals. We appreciate your positive feedback.

Introduction:

Line 43: cite a review in which the different vertebrate and invertebrate hosts are reported.

New references #2-4 were added regarding this sentence. We had based on World Health Organization (2015) statements; however, following your suggestion, the reviews you mentioned were included.

Line 45: cite an article in which ChD is reported as the biggest public-health problems in Latin America.

The reference used to write this sentence: “Chagas disease has been one of the biggest public-health problems in Latin America due to the incapacitating effects and mortality” was that of WHO (2015), where it is mentioned that ChD is one of the biggest public health problems; however, consulting the original reference: Rassi et al. 2010, these authors assert that "ChD is becoming an emerging health problem in non-endemic areas…and is one of the world’s 13 most neglected tropical diseases… and ChD has been a scourge to humanity since antiquity, and continues to be a relevant social and economic problem in many Latin American countries.” So, we consider that the interpretation is adequate. In addition, Moncayo and Silveira (2009) assert that “the impact on the decrease of the burden of disease measured in DALYs (Disability-Adjusted Life Years) lost due to ChD has been the most important in the period 1990-2001 for Latin America and the Caribbean.” On the other hand, in Marin-Neto and Rassi (2009) review it is reported that ECh is the third most important parasitic disease in the world after schistosomiasis and malaria, so it can be inferred that it is one of the most important in Latin America. For all this, new references #5-7 were added, as you suggested.

Material and Method:

I think the Table 1 is very clear, so sub-section 2.3 is useless.

Sub-section 2.3 was deleted as suggested.               

From line 157 to 163: This is results. And it is missing in the result part.

Because these data are not results that have been generated in this article, we consider they could be added in the Materials and Methods section; however, as you are suggesting, they were moved to the results section with their respective reference, mentioning the previous published report.

Line 183 « ] » is missing at the end of the formula.

Thank you for pointing this out. We have checked and corrected the formula.

Statistical analyzes: Did authors test the normality prior to use the One-Way ANOVA test?

Yes, we did. The SPSS Kolmogorov-Smirnov test for normality was used because our sample sizes were N ≥ 25.

If yes, please indicate which test was done. If no, please test the normality and change to non-parametric test instead of One-way ANOVA if the data do not follow a normal distribution.

We have rephrased the sentence to emphasize that our data obtained were continuous variables with a normal distribution so we analyzed them using the one-way ANOVA parametric test followed by Tukey’s analysis.

Results

I would be curious to have correlation tests between the results of echocardiography and the parasitic load. Maybe, the vaccination leaded to parasitic load decrease and so a diminution of the effect on heart? I recommend to discuss about other diseases which could lead to the same echocardiographic profile. They could interfere in the first results and also in the following of the disease. This point should also be discussed in discussion section.

Unfortunately, we did not have data on the parasite load since it was not possible to make a parasitemia curve during the course of the acute stage, only the presence or absence of parasites was recorded (Rodríguez-Morales et al. 2012, Arce-Fonseca et al. 2013); therefore, we cannot perform correlation tests between echocardiography results and parasite load.

Discussion

I recommend to discuss about other diseases which could lead to the same echocardiographic profile. They could interfere in the first results and also in the following of the disease.

Taking your suggestion into account, in the discussion section, on page #10, the following was added:

“Echocardiography has been very useful in prognosis of some canine diseases. For example, heart disease is not well characterized in adult dogs with Duchenne muscular dystrophy, so Guo et al. defined the clinical course of cardiomyopathy by evaluating echocardiography and cardiac magnetic resonance imaging in 24 Golden Retriever dogs. The study showed that dogs with muscular dystrophy have a progressive cardiomyopathy characterized by left ventricular systolic and diastolic dysfunction, left ventricular chamber enlargement, myocardial tension abnormalities, and myocardial fibrosis similar to that seen in dilated cardiomyopathy [ Guo et al, 2019]. The purpose of another study was to investigate the prognostic value of several clinical variables, EKG, and echocardiography in 63 dogs with idiopathic dilated cardiomyopathy; it was concluded that in dogs with dilated cardiomyopathy the restrictive transmitral flow pattern seems to represent a useful prognostic indicator. HF degree, ascites, end-systolic volume index, and LVEF are also useful parameters if an adequate transmitral flow pattern is not recorded [Borgarelli et al, 2006]. Piantedosi's group studied 120 dogs using standard echocardiography to assess the effect of loading conditions changes on LV volumes before and 24 hours after the patent ductus arteriosus occlusion by Amplatzer Canine Duct Occluder. After ductal closure, a significant reduction of all the parameters examined could be seen: LV internal diameter at end-diastole; LV internal diameter at end-systole; end- diastolic and systolic volume; end-diastolic volume index; and FS. Twenty-four hours after closure, the evaluation of the relative percentage difference of the selected echocardiographic parameters showed a significant reduction, higher in the small size breed dogs than in large size ones [Piantedosi et al, 2019]. Although echocardiographic analysis at varying canine diseases shows a diagnosis of dilated cardiomyopathy, epidemiological data will always provide the guideline to the veterinarian in those cases where ChD is suspected; the gold standard in this case is the serological diagnosis.”
